# Lightning Current Measurement Method Using Rogowski Coil Based on Integral Circuit with Low-Frequency Attenuation Feedback

**DOI:** 10.3390/s24154980

**Published:** 2024-08-01

**Authors:** Yiping Xiao, Hongjian Jiao, Feng Huo, Zongtao Shen

**Affiliations:** 1School of Electrical and Electronic Engineering, Hubei University of Technology, Wuhan 430068, China; 102200217@hbut.edu.cn (H.J.);; 2Hubei Key Laboratory for High-Efficiency Utilization of Solar Energy and Operation Control of Energy Storage System, Hubei University of Technology, Wuhan 430068, China; 3China Electric Power Research Institute Co., Ltd., Wuhan 43007, China

**Keywords:** Rogowski coil, lightning current measurement, low-frequency distortion, integral correction, low-frequency attenuation feedback network

## Abstract

A lightning current measurement method using a Rogowski coil based on an integral circuit with low-frequency attenuation feedback was proposed to address the issue of low-frequency distortion in the measurement of lightning currents on transmission lines using Rogowski coils. Firstly, the causes of low-frequency distortion in lightning current measurements using Rogowski coils were analyzed from the perspective of frequency domains. On this basis, an integration correction optimization circuit with a low-frequency attenuation feedback network was designed to correct the low-frequency distortion. The optimized integration circuit can also reduce the impact of low-frequency noise and the DC bias of the operational amplifier (op-amp) on the integration circuit due to the high low-frequency gain. Additionally, a high-pass filtering and voltage-divided sampling circuit has been added to ensure the normal operation of the integrator and improve the measurement range of the measurement system. Then, according to the relationship between the amplitude–frequency characteristics of the measurement system and the parameters of each component, the appropriate types of components and op-amp were selected to expand the measurement bandwidth. Finally, a simulation verification was conducted, and the simulation results show that this measurement method can effectively expand the lower measurement frequency limit to 20 Hz, correct the low-frequency distortion caused by Rogowski coils measuring lightning currents on transmission lines, and accurately restore the measured lightning current waveform.

## 1. Introduction

Most transmission lines are widely distributed in the environment of complex climatic conditions, which means transmission lines very easily suffer from lightning strikes. In recent years, lightning strikes have become the main factor in power system operation failures [1,2]. Obtaining accurate lightning current parameters is not only a prerequisite for studying lightning characteristics, analyzing lightning faults, and locating lightning faults, but it is also a basis for various lightning protection designs for power systems [3,4,5]. Therefore, it is of great significance to accurately measure lightning currents to obtain precise lightning current data.

Lightning current is a powerful current released in the form of pulses within a short period. According to the International Electrotechnical Commission (IEC), the magnitude of the recommended lightning current typically ranges from 7 kA to 52 kA, with the energy spectrum primarily concentrated between a few hundred Hz and several hundred kHz. Taking the standard 8/20 μs lightning current waveform as an example, its frequency band spans from 100 Hz to 500 kHz. To accurately measure the full-wave signal of the lightning current on transmission lines, sensors need to have a wider measurement bandwidth. In the self-integrating operational state, the Rogowski coil does not experience magnetic saturation, has no direct electrical connection with the measured circuit, and exhibits a wide measurement bandwidth and strong anti-interference capability [6,7,8]. Therefore, it is often used for measuring lightning current parameters characterized by a high-amplitude and wide-frequency range [9,10]. Within its self-integrating operational bandwidth, the Rogowski coil features a flat amplitude–frequency response. The output signal of the Rogowski coil is proportional to the measured signal, and it has an upper measurement frequency limit reaching several MHz, which is sufficient to meet the measurement requirements for the high-frequency components of lightning currents. However, due to the constraints of the Rogowski coil’s material, structure, and the interdependence of various parameters, it is difficult to obtain a lower limit measurement frequency below 100 Hz for a Rogowski coil in the self-integrating operational state. When the frequency of signal is below the lower limit measurement frequency, the output signal of the Rogowski coil is the derivative of the measured signal [11,12,13]. Additionally, during the transmission of lightning current through the transmission line, attenuation and distortion occur, increasing the components of the lightning current signal with frequencies lower than the Rogowski coil’s lower measurement frequency limit. This results in a distortion in the Rogowski coil’s measured output of lightning current waveforms on transmission lines [14]. To address this issue, the structural parameters of the Rogowski coil can be designed to reduce the lower measurement frequency limit in the self-integrating operational state. However, this method will also decrease the upper measurement frequency limit and measurement sensitivity to some extent, making it challenging to achieve an optimal parameter design [6,15]. Alternatively, an external integrator can be used to perform an integral correction on the distorted signal output by the Rogowski coil [11,16]. Passive integrators have poor integration characteristics at low frequencies and reduce the system’s measurement sensitivity [17]. In Reference [18], a Rogowski coil combined with a passive RC external-integration-correction circuit was used to reduce the lower measurement frequency limit to 160 Hz. However, this is insufficient to meet the measurement requirements for lightning currents on transmission lines, and its measurement sensitivity is only 20 mV/kA. In References [11,19], both reverse-phase active integrators and in-phase active integrators were used to integrate the derivative output of the Rogowski coil. The reverse-phase active integrators have only one turnover frequency and operate in the integration state only above this frequency, making them unsuitable for integrating the Rogowski coil’s output signal below the lower measurement frequency limit. The in-phase active integrators can integrate input signals within a specific frequency band, and the integration upper and lower limits can be freely adjusted by tuning the resistance and capacitance parameters. This makes them suitable for correcting low-frequency distortion in the Rogowski coil’s measurement of lightning currents. However, the in-phase active integrators introduce a constant low-frequency gain G_d_ for low-frequency signals, and this low-frequency gain G_d_ increases with the expansion of the integration bandwidth. This may cause low-frequency noise and the op-amp’s own DC bias voltage to introduce deviations in the integrators’ outputs and even affect their normal operation. Therefore, further improvements to in-phase active integrators are necessary [20]. Reference [21] introduced a high-pass filter after the in-phase active integrators, resulting in a combined amplitude–frequency characteristic curve where the low-frequency gain decreases as the frequency decreases, but it does not reduce the low-frequency gains of active integrators themselves in the low-frequency range.

In this paper, the reasons for low-frequency distortion in lightning current measurements on transmission lines by Rogowski coils were analyzed, the conventional active integration circuit was optimized, and a method for measuring lightning currents on transmission lines using a Rogowski coil based on an integral correction circuit with low-frequency attenuation feedback was proposed. This measurement method allows for the extension of the lower measurement frequency limit of the Rogowski coil and the correction of low-frequency distortions in Rogowski-coil lightning-current measurements by adjusting the parameters of the components in the designed integrator circuit. Additionally, it also improves the performance of the in-phase active integrators at low frequencies and reduces the impact of low-frequency noise and op-amp’s DC bias on the integrator circuit. The analysis of the measurement principles and final simulation results demonstrates that this method can solve the problem of low-frequency distortion and accurately reproduce the measured lightning current waveform on transmission lines.

## 2. Lightning Current Modeling and Spectral Analysis

According to IEC 60060-1 [22], the 8/20 μs waveform is used as the standard lightning current waveform on transmission lines for equipment testing [18,23,24]. In addition, in engineering applications, the double-exponential model is commonly used to simulate the waveform of 8/20 μs lightning currents on transmission lines, whose main characteristic parameters include the wave front attenuation coefficient α, the wave tail attenuation coefficient *β*, the peak correction factor *η*, and the peak value of the lightning current *I*_0_ [25,26]. The double-exponential model is:(1)$$I(t)=\frac{I_{0}}{\eta }[e^{-\alpha t}-e^{-\beta t}]$$
where $\eta =\left(e^{-\alpha t_{p}}-e^{-\beta t_{p}}\right)$, peak time $t_{p}=\ln\left(\alpha /\beta \right)/\left(\beta -\alpha \right)$.

Based on engineering experience, when fitting an 8/20 μs lightning current waveform using a double-exponential model, the empirical values for *α*, *β*, and *I*_0_ are taken as 77,140, 248,900, and 50 kA, respectively. The lightning current waveform is plotted using MATLAB R2022b, and its spectrum is analyzed by fast Fourier transform. The lightning waveforms and its spectrogram are shown in Figure 1a,b.

From the spectrogram, it can be seen that the energy of the lightning current decreases with the increase in frequency and is mainly distributed in the frequency domain from 100 Hz to 500 kHz. Therefore, the measurement band of the lightning current measurement sensor needs to be in the tens of hertz to several megahertz in order to meet the needs of the lightning current measurement.

## 3. Analysis of Low-Frequency Distortion in Lightning Current Measurements by Rogowski Coils

### 3.1. Analysis of the Measuring Principle of Rogowski Coils

A Rogowski coil is essentially a current transformer, and its equivalent circuit is shown in Figure 2. As shown, *I*(*t*) is the measured current; *e*(*t*) is the induced electromotive force coupled inside the Rogowski coil; *M* is the mutual inductance coefficient between the conductor flowing through the measured current and the coil; *L*, *R*, and *C* are the coil self-inductance, internal resistance, and distributed capacitance, respectively; and *R_t_* is the sampling resistor [27,28,29].

The equivalent circuit equation can be obtained as:(2)$$\{\begin{array}{c} e(t)=M\frac{dI(t)}{dt}=L\frac{di(t)}{dt}+Ri(t)+U_{t}(t)\\ i\left(t\right)=C\frac{dU_{t}\left(t\right)}{dt}+\frac{U_{t}\left(t\right)}{R_{t}} \end{array}$$

Due to the typically small value of the inherent distributed capacitance *C* in Rogowski coils, the current flowing through the capacitor C can be neglected when the value of the sampling resistor *R_t_* is also small. Equation (2) can be simplified as Equation (3):(3)$$M\frac{dI(t)}{dt}=L\frac{di(t)}{dt}+\left(R+R_{t}\right)i(t)$$

If the rate of the change of the measured current is relatively large (e.g., thunder current), then Equation (4) is satisfied and shown as follows:(4)$$L\left(\frac{di\left(t\right)}{dt}\right)\gg \left(R+R_{t}\right)i\left(t\right)$$

According to Equation (4), Equation (3) can be further simplified to:(5)$$i(t)\approx \frac{M}{L}I(t)$$

The output voltage of the Rogowski coil:(6)$$U_{t}\left(t\right)=i\left(t\right)R_{t}=\frac{MR_{t}}{L}I(t)$$
where $U_{t}\left(t\right)$ is proportional to the test current $I\left(t\right)$ and the Rogowski coil operates in a self-integrating state.

The Rogowski coil measurement system is analyzed in the frequency domain, and the transfer function *H*_1_(*s*) can be derived from Equation (2). It can be obtained as:(7)$$H_{1}(s)=\frac{U_{t}\left(s\right)}{I\left(s\right)}=\frac{MR_{t}}{R_{t}+R}\frac{s}{\left(T_{L}s+1)(T_{H}s+1\right)}$$
where $T_{L,H}=1/\left(x_{1}\mp \sqrt{x_{1}^{2}-x_{2}^{2}}\right)$, $x_{1}=\frac{1}{2LC}\left(\frac{L}{R_{t}}+RC\right)$, $x_{2}=\frac{1}{\sqrt{LC}}\left(\sqrt{\frac{R_{t}+R}{R_{t}}}\right)$.

According to *H*_1_(*s*), the amplitude–frequency characteristic curve *L*(*ω*) can be shown in Figure 3.

From Equation (7) and Figure 3, the upper frequency limit *f_H_* and the lower frequency limit *f_L_* of the Rogowski coil operating in the self-integrating state can be obtained and shown as Equations (8) and (9).
(8)$$f_{H}=\frac{1}{2\pi T_{H}}=\frac{1}{2\pi R_{t}C}$$
(9)$$f_{L}=\frac{1}{2\pi T_{L}}=\frac{R+R_{t}}{2\pi L}$$

$\left[f_{L},f_{H}\right]$ is the measurement frequency band of the Rogowski coil operating in the self-integrating state. In order to accurately restore lightning current waveforms on transmission lines, the measurement frequency band needs to be able to take into account the low-frequency component and high-frequency component of the lightning current at the same time. For the upper frequency limit *f_H_*, from Equation (8), when the value of *C* is at the nF level and the value of *R_t_* is small, the value of *f_H_* can reach several tens of megahertz, which is enough to meet the demand for measuring the high-frequency component of the lightning current. As for the lower frequency limit *f_L_*, it can be seen from Equation (9) that the lower measurement frequency limit can be reduced by increasing the value of *L* or decreasing the values of *R* and *R_t_*. However, increasing the value of *L* implies an increase in the number of turns of the coil, which leads to an increase in the values of *R*, and decreasing the values of *R_t_* also leads to a decrease in the sensitivity of the measurement system. Combined with the constraints of the coil material and structure design, it is often difficult to obtain the desired lower measurement frequency limit.

### 3.2. Mechanistic Analysis of the Generation of Low-Frequency Distortion in Lightning Current Measurements

In the actual power grid, the frequency and amplitude of lightning currents on transmission lines are subject to attenuation and distortion due to refraction during transmission and environmental disturbances, which will increase the lower-frequency components of the lightning current’s waveform with a frequency less than the lower limit frequency *f_L_*. When these lower-frequency components no longer satisfy Equations (4) and (5) is no longer valid, and the output waveforms become distorted. As shown in Figure 4, when the Rogowski coil is used to measure the 8/20 μs inrush current, the measured waveform shows obvious distortion at the end of the wave with more low-frequency components.

From the Rogowski coil’s amplitude–frequency characteristics shown in Figure 3, it can also be seen that in the frequency domain range of $1/T_{L}\leq {\omega \leq 1/T}_{H}$, the amplitude–frequency characteristic curve is a straight line, and the output voltage signal is proportional to the measured current. In the frequency domain range of $\omega \leq 1/T_{L}$, the Rogowski coil works in a differential state, and the output voltage signal is the derivative of the measured current, which needs to be corrected by the external integral circuit to restore the measured current signal.

## 4. Integral Correction Method for Lightning Current Measurements by Rogowski Coils

### 4.1. Integral Correction Optimized Circuit Design

According to Section 3.2, it can be seen that in order to accurately measure the complete lightning-current waveform, it is necessary to carry out the external integration correction process for the lightning current signal components within $\omega \leq 1/T_{L}$ measured by the Rogowski coil.

The amplitude–frequency characteristic curve of the active integrator shown in Figure 5a is shown in Figure 5b, and it can be seen that the active integrator can integrate the input signal in the frequency range of $\left[\omega_{L},\omega_{H}\right]$. Additionally, the lower measurement limit $\omega_{L}$ can be adjusted by *R_3_*, which is suitable for integrating and correcting the signal output from the Rogowski coil in the frequency range of $\omega \leq 1/T_{L}$. However, there exists a low-frequency gain *G_d_* for signals with frequencies less than $\omega_{L}$. When $\omega_{L}$ is reduced by increasing *R*_3_, it also leads to a larger *G_d_*. Once *G_d_* is too high, the low-frequency noise, as well as the DC bias voltage of the op-amp itself, will distort or even saturate the output signal of the integrator [20]. In order to expand the lower-measurement frequency limit $\omega_{L}$ while reducing the impact of the DC bias of the op-amp and low-frequency noise, it is necessary to optimize the amplitude–frequency characteristics of the active integrator below $\omega_{L}$ so that the *G_d_* can be reduced as the signal frequency decreases, i.e., the amplitude–frequency characteristics before the optimization shown by the solid line are improved to the amplitude–frequency characteristics after the optimization shown by the dashed line in Figure 5b.

#### 4.1.1. Optimized Design of Integral Correction Circuit

In order to optimize the amplitude–frequency characteristics of the active integrator below $\omega_{L}$ as described in Section 4.1, a low-frequency attenuation feedback network consisting of op-amp *A*_2_ was added between the negative input and the output of the conventional active integrator composed of op-amp *A*_1_, shown as *H*_2_ in Figure 6. According to the ideal op-amp characteristics and the circuit principle, the circuit equation of *H*_2_ can be obtained as:(10)$$\frac{U_{1}}{R_{2}}=C_{1}\frac{d\left(U_{O}-U_{1}\right)}{dt}+\frac{U_{O}-U_{1}}{R_{3}}+\frac{\frac{1}{R_{5}C_{2}}\int U_{O}dt-U_{1}}{R_{4}}$$

The transfer function of Equation (10) is obtained by Laplace transform as:(11)$$H_{2}\left(s\right)=\frac{U_{O}\left(s\right)}{U_{1}\left(s\right)}=\frac{R_{2}R_{3}+R_{2}R_{4}+R_{3}R_{4}}{R_{2}R_{3}R_{4}}\frac{\left(1+\frac{R_{2}R_{3}R_{4}}{R_{2}R_{3}+R_{2}R_{4}+R_{3}R_{4}}C_{1}s\right)R_{5}C_{2}R_{4}s}{1+\frac{R_{5}C_{2}R_{4}s}{R_{3}}+R_{5}C_{2}R_{4}C_{1}s^{2}}$$

If satisfying $R_{5}=R_{4}=2R_{3}$ and $C_{2}=C_{1}$, Equation (11) can be simplified as:(12)$$H_{2}\left(s\right)=\frac{{T_{2}}^{2}}{T_{3}}\frac{\left(1+T_{3}s\right)s}{{\left(1+T_{2}s\right)}^{2}}$$
where $T_{2}=R_{5}C_{2}$, $T_{3}=R_{5}R_{2}C_{2}/\left(R_{5}+3R_{2}\right)$.

According to Equation (12), the amplitude–frequency characteristic curve of the integral correction optimization circuit with low-frequency attenuation feedback is shown in Figure 7. From Figure 7, it can be seen that the lower the frequency of the signal below the lower integration frequency limit $1/T_{2}$, the smaller the value of gain *G_d_*. The integral optimisation circuit with low-frequency attenuating feedback has a stronger ability to cut down the low-frequency noise signals as well as the DC bias.

#### 4.1.2. High-Pass Filtering and Voltage-Divided Sampling Circuit Design

Due to the measured lightning current amplitude being very large, up to tens of kiloamperes, the output voltage signal measured by the Rogowski coil can still be up to tens of volts, which can not meet the operating conditions of the op-amp in the active integrator. Therefore, it is necessary to add a high-pass filtering and voltage-divided sampling circuit between the Rogowski coil and the integral calibration circuit. The circuit is shown as *H*_3_ in Figure 6, whose circuit equations and transfer function expressions are Equation (13) and Equation (14), respectively:(13)$$U_{t}=\frac{1}{C_{0}}\int \frac{U_{1}}{R_{1}}dt+\frac{R_{0}}{R_{1}}U_{1}+U_{1}$$
(14)$$H_{3}\left(s\right)=\frac{U_{1}\left(s\right)}{U_{t}\left(s\right)}=\frac{C_{0}R_{1}s}{1+C_{0}\left(R_{0}+R_{1}\right)s}=\frac{C_{0}R_{1}s}{1+T_{1}s}$$
where $T_{1}=C_{0}\left(R_{0}+R_{1}\right)$.

Its amplitude–frequency characteristic curve is shown in Figure 8, and the sampling ratio of the sampling circuit is $R_{1}/\left(R_{0}+R_{1}\right)$.

### 4.2. Amplitude–Frequency Characteristics of Rogowski Coil Measurement System Based on Integral Correction

As can be seen from Section 4.1, the Rogowski coil output signal *U_t_* is first sampled using a high-pass filtering and voltage-divided sampling circuit, and then the integral correction circuit is used to carry out the integral correction process. The transfer function *H*(*s*) of the measurement system with the integral correction optimization circuitry shown in Figure 6 is:(15)$$H(s)=\frac{MR_{t}}{R_{t}+R}\frac{{T_{2}}^{2}}{T_{3}}\frac{s}{\left(T_{H}s+1\right)\left(1+T_{L}s\right)}\frac{C_{0}R_{1}s}{1+T_{1}s}\frac{\left(1+T_{3}s\right)s}{{\left(1+T_{2}s\right)}^{2}}$$

If the parameters are adjusted to make $T_{3}$ = $T_{L}$, $T_{1}$ = $T_{2}$, then Equation (15) can be rewritten as Equation (16) as follows:(16)$$H(s)=\frac{MR_{t}}{R_{t}+R}\frac{{T_{2}}^{2}}{T_{3}}\frac{s}{\left(T_{H}s+1\right)}\frac{C_{0}R_{1}s}{1+T_{2}s}\frac{s}{{\left(1+T_{2}s\right)}^{2}}$$

From Equation (16), the amplitude–frequency characteristic curves of the Rogowski coil measurement system, which combines a sampling circuit and integral correction optimization circuit, are shown as *H*(*s*) in Figure 9.

As can be seen from Figure 9, in region III, the Rogowski coil operates in the self-integrating state, and the output voltage signal is proportional to the input current signal. While in region II, the integral correction optimization circuit integrates the differential output of the Rogowski coil, and the lower measurement frequency of the system is expanded to $1/T_{2}$.

### 4.3. Parametric Design of Sampling Circuit and Integral Correction Circuit

The structural parameters and electrical parameters of the Rogowski coil used for the lightning current measurements on transmission lines in this paper are shown in Table 1. According to Table 1 and Section 4.2, the parameters of components such as resistors and capacitors in the high-pass filtering and voltage-divided sampling circuit and the integral correction circuit were designed. Firstly, in order to meet the measurement demand of the low-frequency component of the lightning current, it is necessary to expand the lower measurement frequency limit *f_L_* to less than 100 Hz by adjusting $1/T_{2}$. Secondly, it is important to ensure that the measurement system *H*(*s*) has a high measurement sensitivity and can measure lightning current signals with as high an amplitude as possible. Finally, in order to ensure the normal operation of the op-amp, the values of the input and output signals of the integral correction circuit must not exceed the power supply voltage (such as ±15 V) of the selected op-amp. Therefore, the value of the sampling ratio of the sampling circuit needs to be reasonably selected. In accordance with the above principles, the design values of each component’s parameter are shown in Table 2.

Based on the parameters in Table 1 and Table 2, the amplitude–frequency characteristics and phase–frequency characteristics of the measurement system before and after correction are obtained using MATLAB, as shown in Figure 10.

As can be seen in Figure 10, the lower measurement frequency limit *f_L_* of the Rogowski coil is 4426 Hz ($2.78\times 10^{4}\;rad/s$) before correction, and the lower measurement frequency limit of the measurement system is expanded to 20 Hz (126 rad/s) after correction, which meets the design requirement that the lower measurement frequency limit should be lower than 100 Hz. Moreover, the phase–frequency characteristics of the system in the frequency domain range of $100\;Hz\sim 10^{5}\;Hz$ ($628\;rad/s\sim 628\times 10^{3}\;rad$) are significantly improved, and the phase error is obviously reduced. From the data in Table 1 and Table 2, the sampling ratio of the sampling circuit can be calculated to be 1/10. In addition, the measurement sensitivity of the measurement system can be obtained by Equation (16) as follows:(17)$$S=\frac{MR_{t}C_{0}R_{1}}{\left(R_{t}+R\right)T_{2}T_{3}}$$

According to the data in Table 1 and Table 2, the measurement sensitivity *S* is $2.49\times 10^{-4}\;V/A$, so that the measured system amplitude (20 log(S)) in the measurement band in Figure 10 is −72.1 dB.

According to the parameters in Table 1 and Table 2, the transfer function model of this measurement system was constructed using MATLAB Simulink as shown in Figure 11, and the output results are shown in Figure 12. The input signal is selected to be a sinusoidal signal with an amplitude of 10 kA and a frequency of 20 Hz, 100 Hz, 1000 Hz, and 2 MHz, respectively.

As can be seen from Figure 12, when the frequency of the input sinusoidal signal is smaller than the lower measurement frequency *f_L_* of the Rogowski coil before correction, the phase shift of the output waveform compared to the input signal waveform is close to 90°, which can be regarded as the derivative of the input signal. After the integral correction, when the frequency of the input sinusoidal signal is greater than 20 Hz, the phase shift between the output waveform of the measurement system and the input signal is 0°. Furthermore, the sinusoidal signal waveform of the input, which is scaled down in proportion to the measurement sensitivity *S*, coincides exactly with the output waveform after the correction, thus the correctness of the design theory of this measurement system was verified.

### 4.4. Selection of Op-Amps

Whether the integrator works properly depends on whether the upper integration frequency limit $1/T_{3}$ that the integrator can actually achieve is related to the gain bandwidth product (GBW) and the slew rate (SR) of the op-amp used. From H_2_ (s) in Figure 9, it can be seen that the integral correction circuit has a gain of 1 after the upper cut-off frequency, and that the GBW of the op-amp needs to be greater than 1 MHz if lightning currents with a high-frequency component of up to 1 MHz are to be measured. Furthermore, in order to make the output signal of the op-amp distortion-free, the SR of the op-amp also needs to satisfy Expression (18) to make the rate of the change of the input signal of the op-amp always less than the SR of the op-amp.
(18)$$2\pi BW\frac{MR_{t}}{R_{t}+R}\frac{1}{10T_{L}}I_{p}\leq SR$$

If the measured lightning current amplitude *Ip* is 50 kA and the desired bandwidth BW is 2 MHz, the SR needs to satisfy Expression (19) based on the values of *M*, *L*, *R_t_*, and *R* in Table 1 and Table 2.
(19)SR≥156 V/μs

Since the measurement sensitivity *S* of the measurement system is $2.49\times 10^{-4}\;V/A$, the maximum input and output voltages of the op-amp in the integration circuit are
(20)$$U_{\max}=I_{p}\cdot S=12.45\;V$$

Therefore, it is necessary to select an op-amp with GBW≫1 MHz, SR≫156 V/μs and a power supply voltage of ±15 V.

## 5. Simulation Verification

The measurement-system simulation-circuit model is built using Multisim 14.3 software as shown in Figure 13. As shown, S1 is an 8/20 μs inrush current waveform generation circuit with an amplitude of 50 kA, which is used to simulate the lightning current on transmission lines. The values of the wave modulation resistors RS1 and RS2 are 0.5 Ω and 0.001 Ω, respectively. The value of the wave modulation inductor LS1 is 5.48 μH, the value of the energy storage capacitor CS1 is 12 μF, and the charging voltage is 54,500 V. S2, S3, and S4 are the equivalent circuit of the Rogowski coil, the high-pass filtering and voltage-divided sampling circuit, and the integral correction circuit with a low-frequency attenuation feedback network, respectively. Based on the discussion in Section 4.3, the THS4012ID is selected as the op-amp in the integral optimization correction circuit.

According to the above simulation model, the comparison diagram between the 8/20 μs lightning current test waveforms and the measurement system output waveforms before and after correction is shown in Figure 14 and Figure 15.

From Figure 14 and Figure 15, it can be seen that the output waveforms corrected by the integral correction circuit with low-frequency attenuation feedback overlap well with the lightning current test waveforms. This indicates that the designed lightning current measurement method using a Rogowski coil based on the integral circuit with low-frequency attenuation feedback can accurately restore the lightning current generated on the transmission line.

In order to test the accuracy and linearity of this measurement system, different peak input inrush currents were obtained by changing the charging voltage of the energy storage capacitor in the 8/20 μs inrush current waveform generating loop. Additionally, the corresponding peak values of the measured output voltages were recorded in the oscilloscope 2, and the corresponding theoretical output voltage peaks were calculated according to the measurement sensitivity *S* ($2.49\times 10^{-4}\;V/A$) of the measurement system. The measured output voltage peak curve and theoretical output voltage peak curve corresponding to different peak input impulse currents are shown in Figure 16. It can be found that the fitting curve of the measured output value follows a straight line and coincides with the fitting curve of the theoretical output value, which indicates that the system has a very good measurement accuracy in the measurement frequency domain.

## 6. Conclusions

In this paper, based on the analysis of the spectrum of lightning currents on transmission lines and the mechanism of low-frequency distortion generated by measuring lightning currents using a Rogowski coil, a method for measuring lightning currents on transmission lines using a Rogowski coil was proposed. This method is based on an integral correction circuit with low-frequency attenuation feedback, and it has been theoretically analyzed and simulated for verification as follows:
This measurement method optimizes the operating characteristics of the active integrated circuit in the low-frequency band by adding a low-frequency attenuation feedback network to the active integrated circuit, which suppresses the effects of the DC bias of the op-amp and low-frequency noise on the measurement.In order to satisfy the operating conditions of the op-amp in the active integrator circuit, a high-pass filtering voltage-divided sampling circuit was introduced before the integrator correction circuit.From the perspective of the frequency domain, the measurement principle of the entire measurement system was analyzed. The values of each component in each section were designed so that the measurement sensitivity of the measurement system was $2.49\times 10^{-4}\;V/A$. Additionally, the lower measurement frequency limit of the measurement system was extended to 20 Hz, the phase error in the low-frequency range was reduced, and the low-frequency distortion caused by measuring lightning currents on transmission lines using a Rogowski coil was corrected.Finally, according to the simulation results, it can be seen that the method can accurately restore lightning current signals on transmission lines, which is convenient for accurately locating lightning fault locations in later stages. In addition, the design method can also be used to adjust the measurement range by selecting the appropriate integrator circuit parameters according to the measurement object, which can be applied to other high-frequency pulse current measurements.

## Figures and Tables

**Figure 1 sensors-24-04980-f001:**
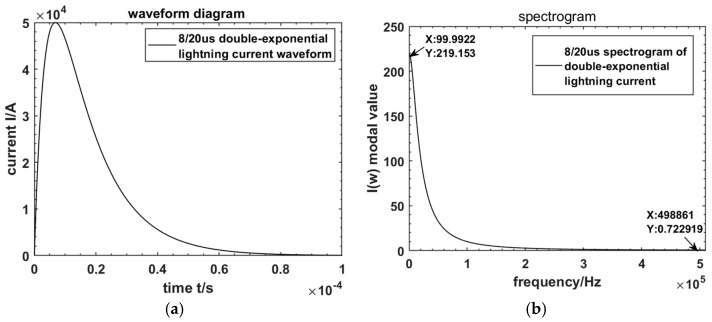
Waveforms and spectrograms of double-exponential lightning current model. (**a**) Lightning current waveform; (**b**) lightning current spectrogram.

**Figure 2 sensors-24-04980-f002:**
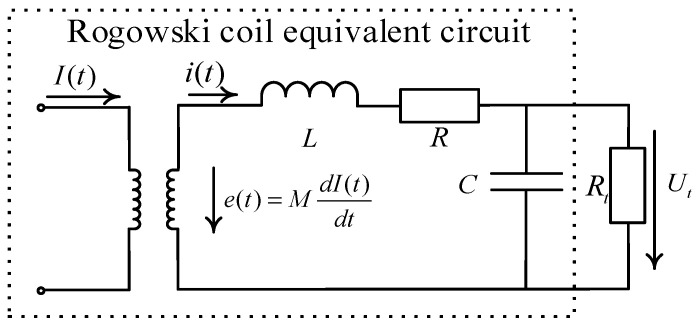
Equivalent circuit diagram of Rogowski coil.

**Figure 3 sensors-24-04980-f003:**
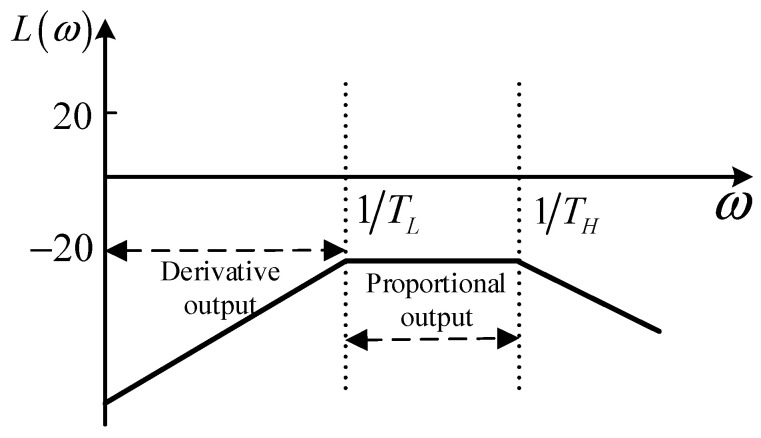
Rogowski coil amplitude–frequency characteristic curve.

**Figure 4 sensors-24-04980-f004:**
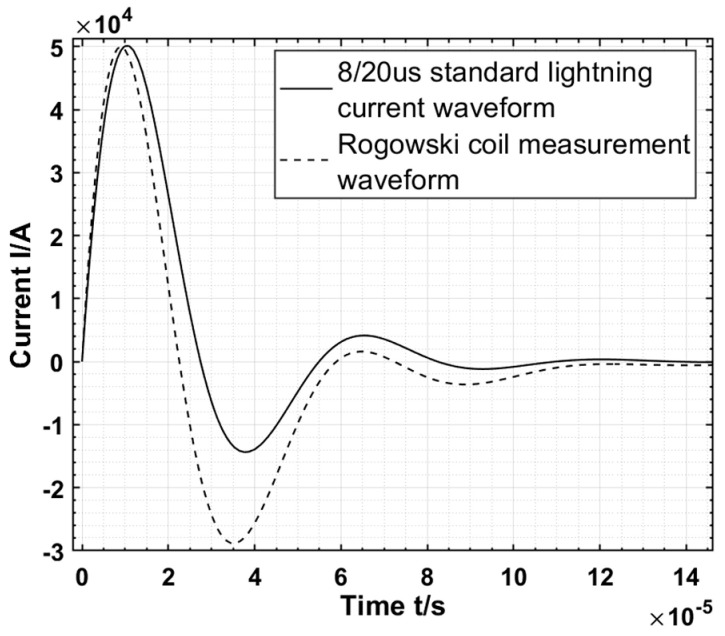
Low−requency distortion diagram.

**Figure 5 sensors-24-04980-f005:**
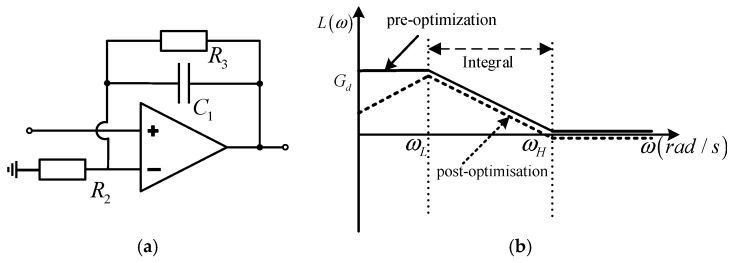
Active integrator and its amplitude–frequency characteristic curves. (**a**) Active integrator; (**b**) the amplitude–frequency characteristic curves of the active integrator.

**Figure 6 sensors-24-04980-f006:**
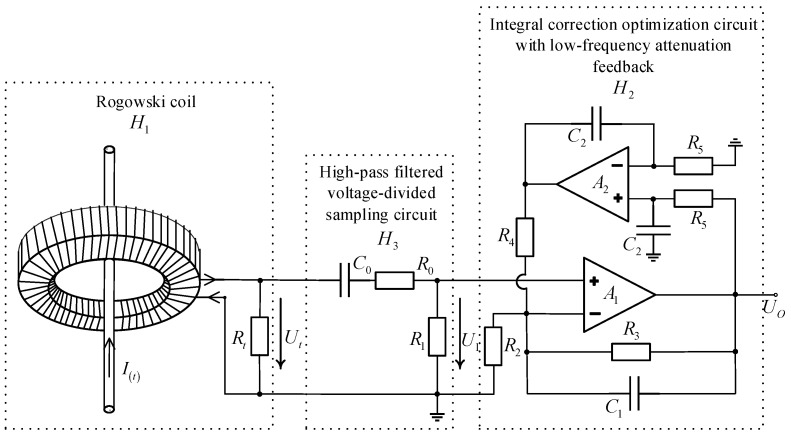
Measurement system with integral correction optimization circuitry.

**Figure 7 sensors-24-04980-f007:**
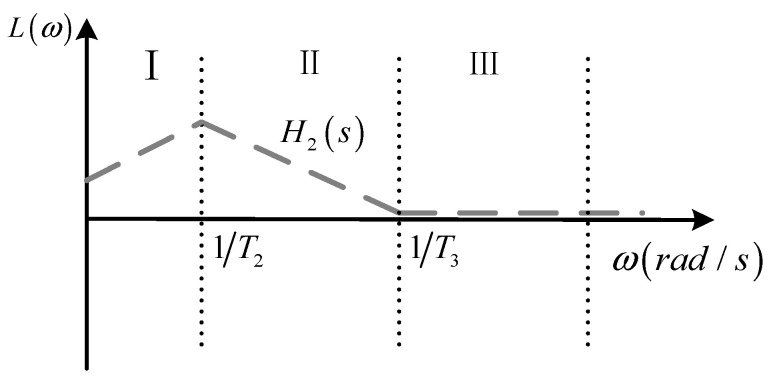
Amplitude–frequency characteristic curve of the integral correction optimization circuit.

**Figure 8 sensors-24-04980-f008:**
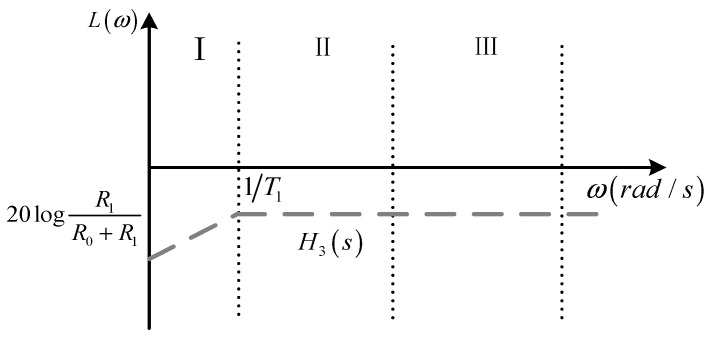
Amplitude–frequency characteristic curve of sampling circuit.

**Figure 9 sensors-24-04980-f009:**
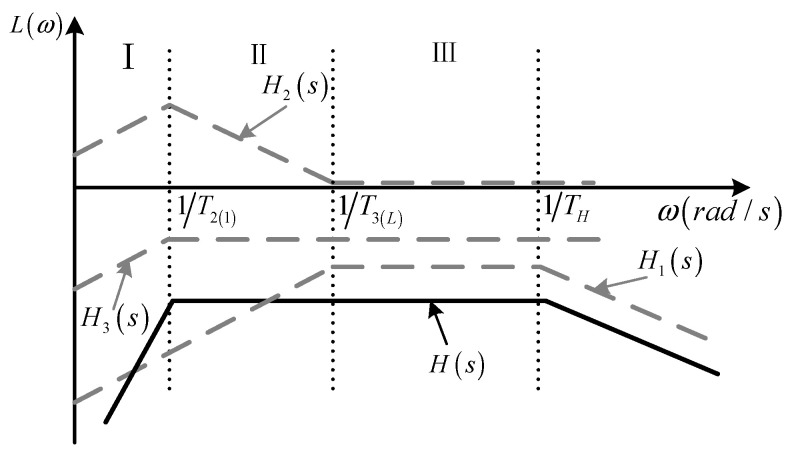
Amplitude–frequency characteristic curves of the measurement system.

**Figure 10 sensors-24-04980-f010:**
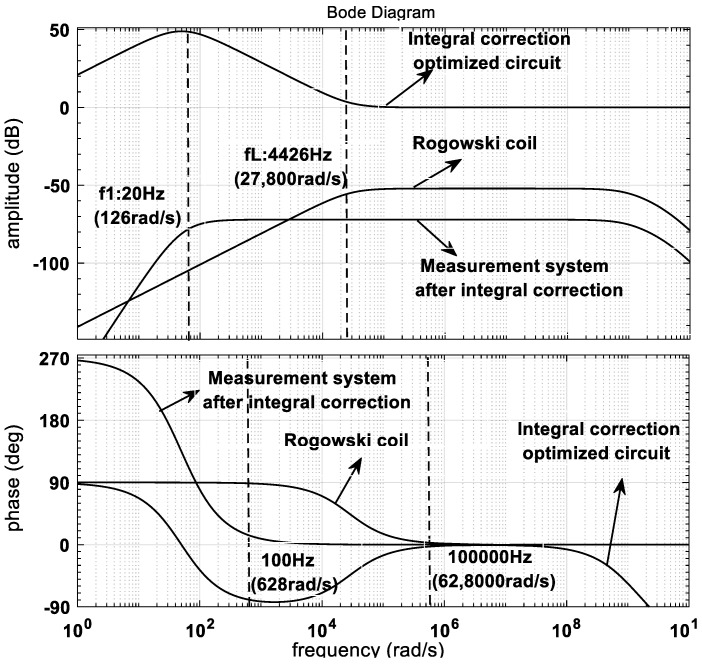
Frequency characteristics of the measurement system before and after correction.

**Figure 11 sensors-24-04980-f011:**
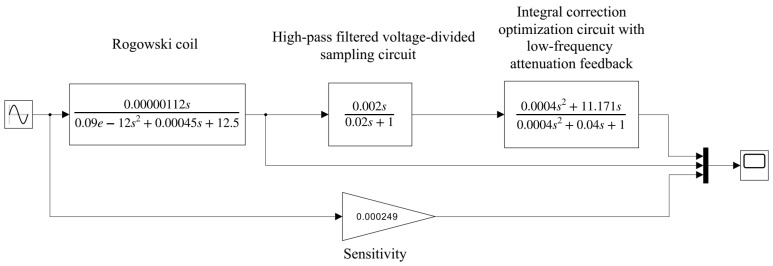
System transfer function model.

**Figure 12 sensors-24-04980-f012:**
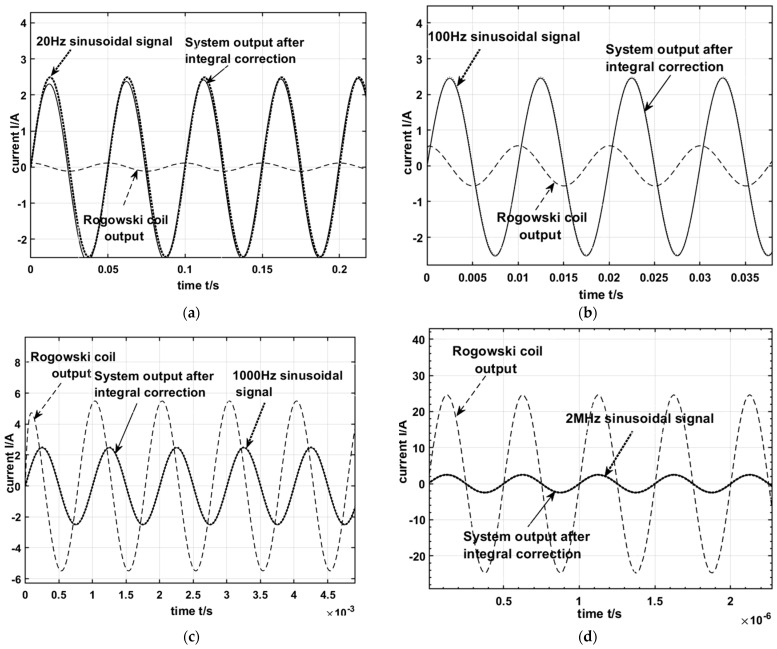
Output results of the system before and after correction when the input signal amplitude is 10 kA and the frequencies are 20 Hz, 100 Hz, 1000 Hz, and 2 MHz, respectively. (**a**) Signal fre–quency: 20 Hz, (**b**) signal frequency: 100 Hz, (**c**) signal frequency: 1000 Hz, and (**d**) signal frequency: 2 MHz.

**Figure 13 sensors-24-04980-f013:**
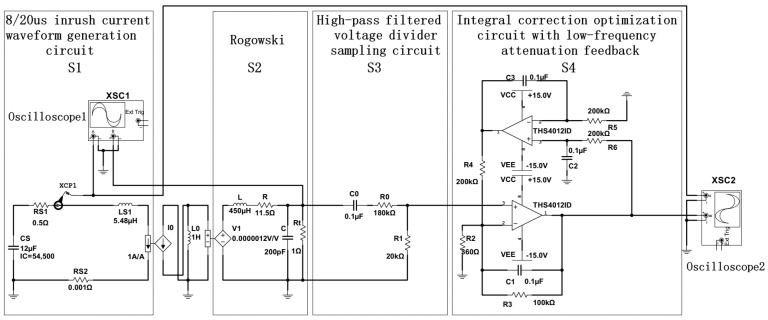
Multisim simulation circuit model.

**Figure 14 sensors-24-04980-f014:**
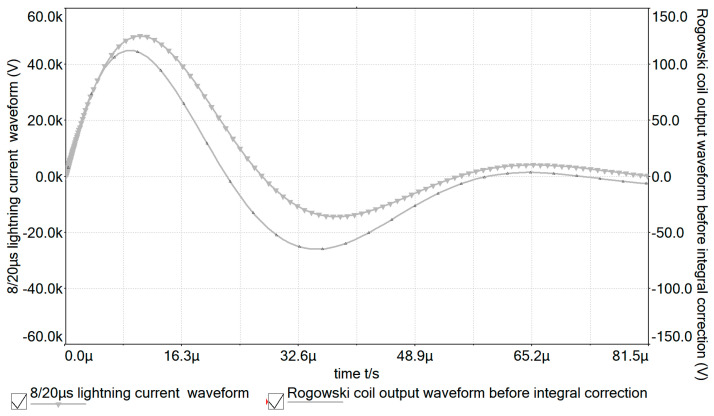
Comparison between output waveform before integration correction and 8/20 μs lightning current test waveform.

**Figure 15 sensors-24-04980-f015:**
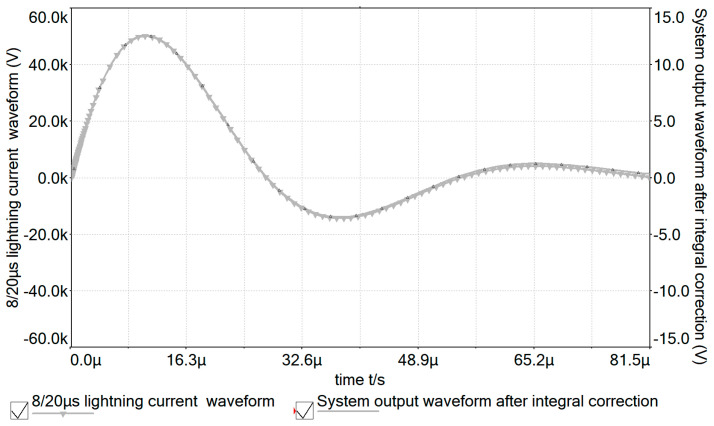
Comparison between output waveform after integration correction and 8/20 μs lightning current test waveform.

**Figure 16 sensors-24-04980-f016:**
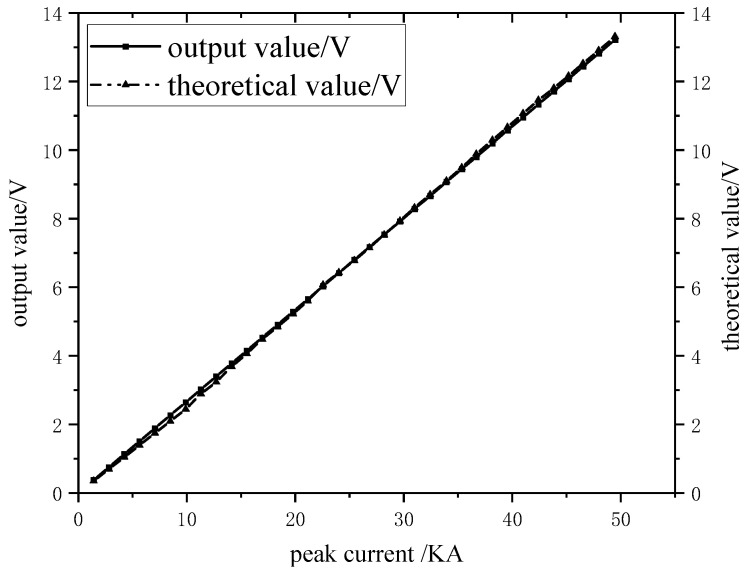
Measured voltage peak curve and theoretical output voltage peak curve.

**Table 1 sensors-24-04980-t001:** Rogowski coil structure and electrical parameters.

Parameters	Value
Turns/N	400
Inside diameter D/m	0.7
External diameter d/m	0.9
Height h/cm	5
Mutual inductance M/μH	1.12
Self-inductance L/mH	0.45
Internal resistance R/Ω	11.5
Distributed capacitance C/nF	0.20
Sampling resistor $R_{t}/\Omega$	1

**Table 2 sensors-24-04980-t002:** Electrical parameters of sampling circuit and integral correction circuit.

*C*_0_/μF	*R*_0_/kΩ	*R*_1_/kΩ	*C*_1_/μF	*R*_2_/kΩ	*R*_3_/kΩ	*R*_4_/kΩ	*C*_2_/μF	*R*_5_/kΩ
0.1	180	20	0.1	360	100	200	0.1	200

## Data Availability

Data are contained within the article.

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
