# Peer review of "Lightning Current Measurement Method Using Rogowski Coil Based on Integral Circuit with Low-Frequency Attenuation Feedback"

_sensors, 2024, doi:10.3390/s24154980_

Round 1

Reviewer 1 Report

Comments and Suggestions for Authors

In this paper an integral circuit with low-frequency attenuation feedback was proposed to address the issue of low-frequency distortion in the measurement of lightning current on transmission lines using Rogowski coils. The effect of the circuit is analyzed theoretically and verified by experiments. The results are credible and valuable.

The formation of the equation can be improved.

The study's originality lies in the design of an integration correction optimization circuit with a low-frequency attenuation feedback network to address low-frequency distortion.

The authors should consider detailing the calibration process for the measurement system and provide more extensive validation through experimental data in addition to simulations.

Further controls could include testing the system under varying environmental conditions and with different types of lightning currents to ensure robustness.

The authors should consider detailing the calibration process for the measurement system and provide more extensive validation through experimental data in addition to simulations.

Further controls could include testing the system under varying environmental conditions and with different types of lightning currents to ensure robustness.

end comments

Reviewer 2 Report

Comments and Suggestions for Authors

The presented article contains significant content for the theme and the methodology is appropriate. However, the contribution of the article lacks clarity. I suggest the following reference [L. Ming, Z. Xin, C. Yin, M. Chen and P. C. Loh, "Integrator Design of the Rogowski Current Sensor for Detecting Fast Switch Current of SiC Devices," 2019 IEEE Energy Conversion Congress and Exposition (ECCE), Baltimore, MD, USA, 2019, pp. 4551-4557] in the literature review for comparison with the proposed work.

Other points that should be considered:

  • Section 1, paragraph 2: Comment on, or at least reference, the necessary requirements (in values) for  lightning current measurements (approximate frequencies and magnitudes);
  • Section 1, paragraph 2: Define what constitutes low/poor performance with specific values to avoid subjectivity;
  • Section 1, paragraph 3: Remove '...the method is simple and feasible...', this statement should be implicitly understood by the reader;
  • Section 2: Better explain the 8/20 μs parameter and reference the standard/regulation. The references currently in the text are articles, not standards/regulations. If none exist, then state that this is a consensus value adopted;
  • Better explain the coefficients alpha, beta, and tp, and how they are obtained or estimated;
  • Perform a detailed review of the text, paying attention to spacing, sentence beginnings, and punctuation.
Comments on the Quality of English Language

The text must be reviewed for punctuation, spaces and lowercase letters at the beginning of sentences.

Author Response

Comments 1: The presented article contains significant content for the theme and the methodology is appropriate. However, the contribution of the article lacks clarity. I suggest the following reference [L. Ming, Z. Xin, C. Yin, M. Chen and P. C. Loh, "Integrator Design of the Rogowski Current Sensor for Detecting Fast Switch Current of SiC Devices," 2019 IEEE Energy Conversion Congress and Exposition (ECCE), Baltimore, MD, USA, 2019, pp. 4551-4557] in the literature review for comparison with the proposed work.

Response 1: The reference [L. Ming, Z. Xin, C. Yin, M. Chen and P. C. Loh, “Integrator Design of the Rogowski Current Sensor for Detecting Fast Switch Current Current Devices ,” 2019 IEEE Energy Conversion Congress and Exposition (ECCE), Baltimore, MD, USA, 2019, pp. 4551-4557] was introduced at the end of the second paragraph of section 1, and analysis of its measurement method was conducted. The specific modifications are highlighted in green background at the end of the second paragraph of section 1 in the revised manuscript.

Comments 2:Section 1, paragraph 2: Comment on, or at least reference, the necessary requirements (in values) for  lightning current measurements (approximate frequencies and magnitudes);

Response 2: Approximate values of frequencies and magnitudes of lightning current measurements was given at the second paragraph of section 1. The specific modifications are highlighted in green background and approximate values are highlighted in green font in section 1, paragraph 2 in the revised manuscript .

Comments 3: Section 1, paragraph 2: Define what constitutes low/poor performance with specific values to avoid subjectivity;

Response 3: The description of the specific values of the poor performance was added in Section 1, paragraph 2 in the revised manuscript and marked in green background in the first three places of the second paragraph of section 1, specific values are highlighted in green font.

Comments 4: Section 1, paragraph 3: Remove '...the method is simple and feasible...', this statement should be implicitly understood by the reader;

Response 4: In the revised manuscript, NOT ONLY the sentence '...the method is simple and feasible...' was removed, BUT ALSO the paragraph was rewritten to make the expression clearer. The specific modifications are highlighted in green background in Section 1, paragraph 3 in the revised manuscript.

Comments 5: Section 2: Better explain the 8/20 μs parameter and reference the standard/regulation. The references currently in the text are articles, not standards/regulations. If none exist, then state that this is a consensus value adopted;

Response 5: The 8/20 μs parameter comes from standard IEC 60060-1. According to IEC 60060-1, the 8/20 μs exponential waveform is used as the standard lightning current waveform for equipment testing. The standard has been added in section 2, and information about this standard also has been added in references in the revised manuscript. The modifications highlighted in green background in section 2 of the revised manuscript .

Comments 6: Better explain the coefficients alpha, beta, and tp, and how they are obtained or estimated;

Response 6: The coefficients alpha, beta, and tp are wave front attenuation coefficient, the wave tail attenuation coefficient, the peak time respectively. And these coefficients are empirical values or consensus values , which was obtained through data fitting using a program.

Comments 7: Perform a detailed review of the text, paying attention to spacing, sentence beginnings, and punctuation.

Response 7: In the revised manuscript, spacing, sentence beginnings, and punctuation in this paper have been thoroughly rechecked and revised, and the modifications to individual contents in the article have also been highlighted in green in the revised manuscript .

Round 2

Reviewer 2 Report

Comments and Suggestions for Authors

The authors addressed the issues raised in the first round of review